# BIVLC: Extending Vision-Language Compositionality Evaluation with Text-to-Image Retrieval

**Imanol Miranda    Ander Salaberria    Eneko Agirre    Gorka Azkune**
HiTZ Center – Ixa, University of the Basque Country (UPV/EHU)
{imanol.miranda, ander.salaberria, e.agirre, gorka.azcune}@ehu.eus

## Abstract

Existing Vision-Language Compositionality (VLC) benchmarks like SUGAR-CREPE are formulated as image-to-text retrieval problems, where, given an image, the models need to select between the correct textual description and a synthetic hard negative text. In this work, we present the Bidirectional Vision-Language Compositionality (BIVLC) dataset. The novelty of BIVLC is to add a synthetic hard negative image generated from the synthetic text, resulting in two image-to-text retrieval examples (one for each image) and, more importantly, two text-to-image retrieval examples (one for each text). Human annotators filter out ill-formed examples ensuring the validity of the benchmark. The experiments on BIVLC uncover a weakness of current multimodal models, as they perform poorly in the text-to-image direction. In fact, when considering both retrieval directions, the conclusions obtained in previous works change significantly. In addition to the benchmark, we show that a contrastive model trained using synthetic images and texts significantly improves over the base model in SUGARCREPE and in BIVLC for both retrieval directions. The gap to human performance in BIVLC confirms that Vision-Language Compositionality is still a challenging problem. BIVLC and code are available at https://imirandam.github.io/BiVLC_project_page.

## 1   Introduction

Vision-Language Compositionality (VLC) refers to the ability to discern whether the composition of a given set of elements is the same for an image and a text. For example, assuming an image depicts a black dog and a white cat, but the text states that there is a white dog and a black cat, both compositions do not match. VLC performance is usually evaluated on image-to-text retrieval datasets[Hsieh et al., 2024, Ma et al., 2023, Yuksekgonul et al., 2022]: given an image and some textual descriptions, a model has to retrieve the description which matches the image. The candidates include hard negative distractors which differ from the correct description only in compositional details. However, there is no theoretical reason to favour image-to-text retrieval over text-to-image retrieval. The dominant use of the former lies probably on practical reasons, i.e. it is much easier to collect hard negative texts than hard negative images.

In this paper, we propose a semi-automatic method to generate an evaluation dataset for VLC that includes image-to-text and text-to-image retrieval (referred as I2T and T2I respectively). We refer to this problem *bidirectional* Vision-Language Compositionality, since it combines both retrieval directions. For that purpose, we extend the SUGARCREPE dataset [Hsieh et al., 2024], the reference benchmark for image-to-text retrieval-based VLC. We leverage the hard negative textual descriptions provided in the dataset and, for every hard negative text, we generate four candidate images using off-the-shelf text-to-image generators. We ask human annotators to select the image that better describes the text and to discard those instances where no image is correct. In a second annotation step, we ask annotators to remove instances where the generated image matches both captions,

38th Conference on Neural Information Processing Systems (NeurIPS 2024) Track on Datasets and Benchmarks.

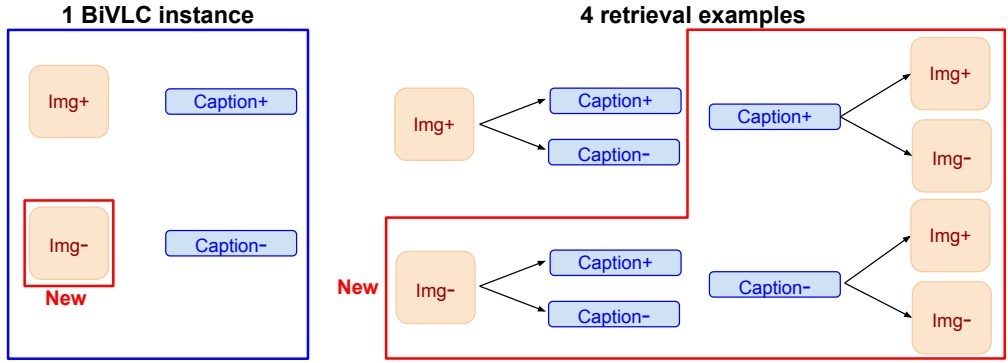

Figure 1: Given an image and two captions from SUGARCREPE, BIVLC constructs an instance adding a negative image (Img-) generated from the negative caption (Caption-). The instance produces four retrieval examples: two for image-to-text retrieval and two for text-to-image retrieval.

resulting in an ambiguous instance. As a result, we build a high-quality evaluation dataset called BIVLC (**Bi**directional **V**ision-**L**anguage **C**ompositionality), with almost three thousand instances composed by two images and two captions, which accounts to more than eleven thousand retrieval instances (Figure 1). In addition to the human filtering, we performed experiments which show that distinguishing between synthetic and natural images-texts is not enough to perform well in BIVLC, confirming the validity of the synthetic images.

BIVLC allows us to evaluate models on both text-to-image and image-to-text retrieval, offering a more complete picture of VLC capabilities. The main contribution of this work is the dataset itself, as well as our findings based on the experiments and analysis on BIVLC. We summarize those findings as follows: (i) Humans perform comparably for both retrieval directions, but multimodal models are significantly worse for text-to-image retrieval; (ii) Bidirectional VLC is more difficult than image-to-text retrieval; (iii) The strongest models for image-to-text retrieval are not necessarily the strongest for bidirectional VLC; (iv) Training with hard negative images can boost the performance of multimodal contrastive models, obtaining competitive results for BIVLC, but still far from humans.

## 2 Related Work

To measure Vision-Language Compositionality, image-to-text retrieval is the dominant paradigm in the literature. Different datasets and benchmarks have been proposed following this approach: for example, [Ma et al., 2023] introduced CREPE, a large dataset containing hard negative texts generated by means of heuristic rules. The dataset was designed to measure systematicity and productivity, two important aspects of compositionality. A similar benchmark is ARO [Yuksekgonul et al., 2022], which stands for Attribution, Relation, and Order, and also uses heuristic rules to produce hard negative captions given a matching image and text. ARO also stands out for its size, with more than 50,000 test cases.

Due to the use of heuristic rules for hard negative text generation, [Hsieh et al., 2024] found that very high accuracies can be obtained for those two datasets, even without using the images at all. Taking advantage of the biases introduced by the rule-based hard negative texts, such as nonsensical phrases or grammatically incorrect texts, purely textual models outperform the best multimodal models. Similar language biases have also been studied for concurrent benchmarks by [Lin et al., Tschannen et al., 2024]. In consequence, [Hsieh et al., 2024] proposed a new methodology to produce hard negative texts, leveraging modern Large Language Models (LLM), adversarial refinement and human annotation. As a result, the SUGARCREPE dataset was proposed, which showed that some of the conclusions obtained when using previous datasets, do not hold anymore. In this work, we extend SUGARCREPE with hard negative images, i.e. with images that match the hard negative textual captions and differ from the original caption and image in compositional details. This way, we can measure bidirectional compositionality, i.e. we can measure the performance of multimodal models for image-to-text retrieval and also for text-to-image retrieval.

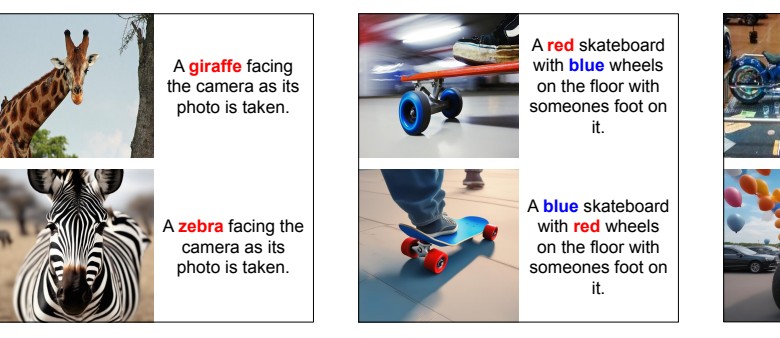

Figure 2: Three instances of BIVLC. Bottom row with negative captions and the corresponding images created by us. From left to right, negative captions created by REPLACE, SWAP and ADD.

In a similar fashion, Winoground [Thrush et al., 2022] also includes both retrieval directions to measure model performance. However, as shown by [Diwan et al., 2022], Winoground has several problems: (i) it is very small (it contains 400 instances), which makes the comparison among models difficult, (ii) it contains very few instances which actually measure Vision-Language Compositionality (only 171), and (iii) some other challenges like commonsense reasoning or locating small, out-of-focus objects in low resolution images are very important to perform well on the task. In this work, mimicking Winoground, we also build every instance of the dataset with two images and two captions, but we only focus on compositionality and we provide almost 3 thousand instances, increasing the size of Winoground considerably.

Recently, two other bidirectional datasets have been published: EqBen [Wang et al., 2023b] and Cola [Ray et al., 2024]. EqBen has been derived from video-text datasets and also offers a set of synthetic images generated with a graphic engine. However, the test set contains low-quality images (e.g. very dark or blurry pictures) [Lin et al., 2024] and authors have released a higher quality subset of 280 pairs of image-text pairs, which is even smaller than Winoground. Cola, on the other hand, has no natural texts, as the captions are produced using templates, and covers only a small subset of compositionality phenomena, as it focuses on object-attributes and spatial relations.

## 3   BIVLC: Bidirectional Vision-Language Compositionality Dataset

We introduce BIVLC, a new benchmark to measure bidirectional vision-language compositionality. Each instance of our dataset is comprised by two images and two captions, with the first pair labeled as positive (image+ and caption+) and the second pair labeled as negative (image- and caption-). For each of the images, one of the captions is the correct one and the other is a hard negative distractor. The same happens for each of the captions. Thus, we can measure image-to-text and text-to-image retrieval with hard negative pairs. Some examples are depicted in Figure 2 (See Appendix G for more examples).

### 3.1   Dataset construction methodology

To generate BIVLC, we take SUGARCREPE as the basis, since it already contains around 7 thousand images with their corresponding positive captions and hard negative captions, carefully created and filtered to avoid any undesirable biases. We apply the following steps (see Figure 3):

**Step 1 - Uniformly format positive and hard negative captions:**   Since the formatting of the captions provided in COCO (which are used also for SUGARCREPE) can vary, starting in lower case, all capitals, missing the final punctuation and so on, we have taken two steps to unify the formatting with the generated captions: 1) process all captions so that they start with capitals and continue in lower case, preserving capitals for named entities, and 2) add the final punctuation mark in case it is missing. This way, we avoid any form-based bias between natural and synthetic captions.

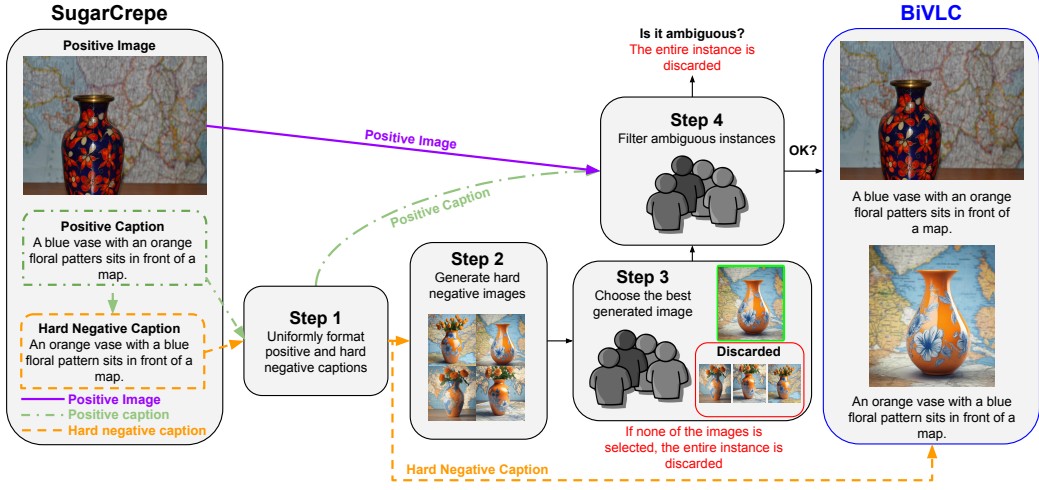

Figure 3: Diagram of dataset construction: Starting from SUGARCREPE instances, uniformly format positive and hard negative captions (Step 1), generate hard negative images (Step 2), ask human annotators to choose the best generated image (Step 3), and filter out ambiguous instances (Step 4). As a result. we get BIVLC instances, consisting of 2 captions and 2 images.

**Step 2 - Generate hard negative images:**  For every hard negative caption in the dataset, we generate 4 corresponding images. For that purpose we use SDXL-DPO [Wallace et al., 2023], a state-of-the-art text-to-image generator, which fine-tunes a Stable Diffusion XL model using the Direct Preference Optimization algorithm [Rafailov et al., 2024] on the Pick-a-Pic dataset [Kirstain et al., 2024]. SDXL-DPO has shown to follow better human preferences than previous models. We basically prompt SDXL-DPO with the hard negative captions and make it generate 4 images, which we store.

**Step 3 - Ask human annotators to choose the best generated image:**  Given a hard negative caption from SUGARCREPE and the 4 images generated in Step 2, we hired human annotators to choose the best of the images. SDXL-DPO does not always generate a suitable image for a given textual description, so annotators were allowed to discard all 4 images. In such cases, we also discard the hard negative caption and its associated positive caption and image (more details about the annotation process in Appendix H.1)). After this annotation step, we have 5,382 instances of two images and two captions. In this step, we used 10 annotators divided into pairs to score a total of 500 annotations and obtain inter-tagger agreement. We calculated Cohen's Kappa score for each pair (Appendix H.3). Since there can be more than one valid images for a caption, we focused on the agreement between selecting any of the 4 images or none. We obtained a mean score of 0.49 between the 5 groups of annotators, which is interpreted as moderate agreement.

**Step 4 - Filter ambiguous instances:**  During Step 3, annotators only see the hard negative caption and the four generated images. As they do not see the original image and its caption, there can be cases where the annotators marked an image as suitable for a given caption, but that same image could also match the original caption, creating an ambiguous instance. This is specially problematic for the ADD category, for which by definition, the generated image also follows the original caption. To minimize those ambiguous instances, we run a filtering step, hiring again human annotators. More concretely, for each instance, we give annotators the original caption and both images. We ask them to choose the image that best describes the caption. If the annotator chooses the generated image, we consider that ambiguous and remove the instance from the dataset. If both images are equally suitable, annotators can mark those instances as ambiguous and we also filter them out.

As a result of this process, BIVLC has 2,933 instances of two images and two captions, or equivalently 11,732 retrieval instances, 50% of which are image-to-text retrieval and the other 50% text-to-image retrieval tests. As we extended SUGARCREPE, we also keep its categories and variants. During both annotation steps (steps 3 and 4) we removed 61% of the instances, which accounts to the noise

Table 1: Number of retrieval examples in VLC datasets, divided into different categories (see text) and total. † Winoground subset with vision-language compositionality is lower, only 684. Winoground has 104 retrieval examples categorized both as SWAP-OBJ and SWAP-REL.

| Dataset | I2T | T2I | REPLACE | | | SWAP | | | ADD | | Total |
|---|---|---|---|---|---|---|---|---|---|---|---|
| | | | OBJ | ATT | REL | OBJ | ATT | REL | OBJ | ATT | |
| Winoground | ✓ | ✓ | | | | 668 | | 1,036 | | | 1,600† |
| SUGARCREPE | ✓ | | 1,652 | 788 | 1,406 | 246 | 666 | | 2,062 | 692 | 7,512 |
| BIVLC (ours) | ✓ | ✓ | 4,800 | 1,748 | 1,848 | 324 | 1,112 | | 1,596 | 304 | 11,732 |

introduced by the text-to-image generator. The statistics of our dataset can be found in Table 1, where we also add SUGARCREPE and Winoground for reference.

## 3.2 Evaluation metrics

The main task defined in BIVLC consists on, given two images and two captions, selecting which image is paired with which caption. To measure the performance on the task, we use the same metrics as Winoground [Thrush et al., 2022], namely, the text, image and group scores. As shown by [Lin et al.], those evaluation metrics discourage blind solutions and are more robust to language biases. For clarity and coherence, we rename the text score as image-to-text accuracy (I2T for short) and the image score as text-to-image accuracy (T2I). Basically, **I2T** measures the performance for image-to-text retrieval, thus it is equivalent to SUGARCREPE. However, in contrast with SUGARCREPE, for each instance in our dataset, we actually have two image-to-text retrieval examples. To obtain a perfect I2T score, the correct captions for both images have to be selected. The **T2I** is similarly defined for text-to-image retrieval. Finally, the **group score** is the main metric, since it combines the performance for image-to-text and text-to-image retrieval. To obtain a perfect group score for a given instance, both images have to be matched with the suitable captions and both captions with the suitable images (formal definitions can be found in Appendix D).

## 4 Experiments and Findings in BIVLC

In this section, we show the experiments performed in BIVLC and our findings using state-of-the-art open multimodal models for VLC. More concretely, we select the following contrastive models: (i) the original CLIP model [Radford et al., 2021], trained on a private dataset of 400M image-caption pairs; (ii) CLIP$_{COCO}$, a CLIP model we fine-tuned on the COCO dataset [Lin et al., 2014] using in-batch negatives (training details in Appendix E); (iii) NEGCLIP, another CLIP model fine-tuned on the COCO dataset augmented with rule-based hard negative captions [Yuksekgonul et al., 2022]; (iv) Generative Negative Mining (GNM) [Sahin et al., 2024], a CLIP model fine-tuned on a dataset which contains hard negative texts and images derived from COCO using image editing techniques. For all of them, we use ViT-B/32 as the backbone visual model, to ensure fair comparisons (cf Appendix E). We also evaluate the following generative models: (i) Open CapPa[1], a recent open-source implementation of CapPa [Tschannen et al., 2024] which has shown similar performance as the original model, based on an encoder-decoder architecture with ViT-L-16 with registers and a decoder of 12 layers; (ii) VQAScore [Lin et al., 2024], based on an architecture which combines a CLIP vision encoder (ViT-L-336) with a Flan-T5 LLM [Chung et al., 2024] of varying sizes (XL refers to 3B and XXL to 11B). Table 2 shows the number of parameters of the different models, with VQAscore being significantly larger.

We also estimate human performance on BIVLC. For that purpose, we hired again human annotators. We took a random sample of 500 instances, and randomly divided them into groups of 25 for each of the 20 annotators. Then, we generated the 4 possible combinations, i.e. image-to-text and text-to-image retrieval, obtaining 2,000 queries. Each query consists of a base text or image and its positive and negative pair of the other modality. The goal is to select the image or text that best represents the base. In consequence, human annotators had to solve independent image-to-text and text-to-image

---

[1]Code and models in https://wandb.ai/craiyon/cappa-jax/reports/CapPa-Training-vision-models-as-captioners—Vmlldzo4NDUyNDUz

Table 2: Results for existing models (see text for details) on SUGARCREPE and BIVLC. For the later, we provide the three metrics I2T, T2I and Group score. Human and random performance are also depicted. Bold for the best of each model family (contrastive and generative).

| | Model | Params | SUGARCREPE | BIVLC I2T | T2I | Group |
|---|---|---|---|---|---|---|
| | Human | N/A | 98.93 | 90.40 | 93.00 | 86.80 |
| | Random | N/A | 50.00 | 25.00 | 25.00 | 16.67 |
| Contrastive | CLIP | | 76.56 | 75.83 | 52.40 | 49.06 |
| | CLIP$_{COCO}$ | 151M | 84.66 | **82.75** | **63.89** | **60.96** |
| | NEGCLIP | | **85.64** | 80.74 | 61.95 | 58.75 |
| | GNM | | 81.83 | 81.32 | 60.86 | 57.96 |
| Generative | Open CapPa | 676M | 90.59 | 57.72 | 56.19 | 41.97 |
| | VQAScore-XL | 3B | 90.85 | 81.96 | 76.61 | 70.20 |
| | VQAScore-XXL | 11B | **93.72** | **86.16** | **81.93** | **76.47** |

retrieval tasks, similarly to how we evaluate multimodal models. Combining the different annotations derived from an instance, we obtain human performance on our three metrics.

Table 2 shows the results of multimodal models both for BIVLC (depicting the three metrics) and for SUGARCREPE. We also show human performance and the scores of a random system. Note that random performance for BIVLC is lower than for SUGARCREPE, which means the task is inherently more difficult.

**Finding 1: Contrary to humans, current models underperform on text-to-image retrieval.** While humans show a balanced performance between text-to-image and image-to-text retrieval scores, the models behave very differently, i.e. the T2I score is in all cases significantly lower than the I2T score, being Open CapPa the only exception but with very low performance for both metrics. Those results show that current multimodal models do not have human-comparable VLC skills at all, and highlight the importance of measuring text-to-image retrieval.

**Finding 2: The gap to humans is bigger in BIVLC than in SUGARCREPE.** The gap between the best model and humans in BIVLC is 10 points (based on group score, since it is the metric that measures bidirectional performance); for SUGARCREPE, 5 points. This means that the bidirectional task is comparatively more difficult than image-to-text retrieval for models than for humans and that there is more room for improvement.

**Finding 3: SUGARCREPE and BIVLC performance are not correlated.** The rank of the multimodal models is different in each of the datasets. For contrastive models, while NEGCLIP is the best model for SUGARCREPE, the best model in BIVLC is CLIP$_{COCO}$, which uses COCO training examples as in-batch negatives. This is surprising: while NEGCLIP uses hard negative texts for training and GNM also adds hard negative images (both derived from COCO), CLIP$_{COCO}$ does not use any technique to mine any kind of hard negatives. On the other hand, the generative Open CapPa model is clearly ahead of all contrastive models for SUGARCREPE, but it is the worst model in BIVLC. This can be explained by the language bias of its LLM decoder, which can be beneficial in SUGARCREPE, but is strongly penalized in our bidirectional setting, as already shown in [Lin et al.].

## 5 Exploring training strategies for vision-language compositionality

We will focus on contrastive models for this exploration, since they are the *de-facto* standard for retrieval problems and are more manageable for training (state-of-the-art generative models are significantly larger). There are two main strategies in the literature to improve the VLC skills of a contrastive model: (i) using hard negative texts for training (exemplified by NEGCLIP [Yuksekgonul et al., 2022]), and (ii) using both, hard negative texts and images (the case of GNM [Sahin et al., 2024]). None of them has been successful in BIVLC. In Sections 5.1 and 5.2 we analyse the

limitations of current implementations and propose two new models based on those strategies. We then evaluate them in SUGARCREPE and BIVLC (Section 5.3).

## 5.1 TROHN-TEXT: Training on Hard Negative Texts

NEGCLIP uses manually defined rules to generate hard negative texts. [Hsieh et al., 2024] showed that texts generated that way show clear biases which prevent models from learning VLC skills properly. To overcome those limitations, we build TROHN-TEXT (**TR**aining **O**n **H**ard **N**egative **Texts**), an automatic large-scale dataset of images and captions. Instead of relying on heuristic rules to mine those hard negative captions as previous works [Ma et al., 2023, Yuksekgonul et al., 2022], we leverage modern LLMs. Basically, we take image-caption pairs from the COCO train split [Lin et al., 2014] and for every caption, we generate as many hard negative captions as category variants defined in SUGARCREPE.

We used the open-source LLM OPENCHAT-3.5-0106 [Wang et al., 2023a] and the templates provided in SUGARCREPE (more details in Appendix F). We generated 591,753 negative captions for each category, for a total of 4,142,271. We then defined a heuristic to filter out the generations that do not follow the templates. After this filtering, we have a total of 3,652,846 negative captions. Finally, we divided the instances consisting of two captions and one image randomly into 80% for training and 20% for validation (more details can be found in Appendix C). We fine-tune a pretrained CLIP model on TROHN-TEXT and call it CLIP$_{\text{TROHN-TEXT}}$ (training details in Appendix E).

## 5.2 TROHN-IMG: Training on Hard Negative Images

GNM [Sahin et al., 2024] used automatically generated images as hard negative examples, but to generate them, GNM uses image editing techniques and is limited to the REPLACE category. SyViC [Cascante-Bonilla et al., 2023] also explores training with synthetic images, but their images are generated with a graphic engine and require techniques such as style transfer to make them more natural.

To overcome those limitations, we design a new dataset named TROHN-IMG (**TR**aining **O**n **H**ard **N**egative **I**mages). This dataset is based on the negative captions previously generated for TROHN-TEXT. Due to the resources and time needed for image generation, TROHN-TEXT is too big (>3M hard negative captions), so we decided to filter the instances to keep the best ones and obtain a training set equivalent to COCO in size [Lin et al., 2014]. This way, we can also measure better the contribution of the hard negative images, since the size of the training set is similar to the one used for our CLIP$_{\text{COCO}}$ model, the best contrastive model for BIVLC so far. Thus, we apply the following steps to generate TROHN-IMG:

**Step 1: Select the best instances based on plausibility and linguistic acceptability:**   Inspired by SUGARCREPE, we used the VERA [Liu et al., 2023] and COLA [Morris et al., 2020] models to score the negative captions and then select instances in the top 35 percentile, according to the combined score of both models. This step also contributes to generate more natural images, since the most implausible captions are discarded.

**Step 2: Data deduplication and cleaning:**   We remove duplicate negative captions, captions in the form of questions, and generations that do not end with a final punctuation mark.

**Step 3: Debiasing the dataset:**   As proposed in SUGARCREPE [Hsieh et al., 2024], we applied adversarial refinement to the remaining captions based on the previous plausibility and linguistic acceptability scores of the positive and negative captions.

**Step 4: Generating images based in negative captions:**   Using the captions obtained in step 3 as prompts, we generate one image per caption with the SDXL-DPO model.

After all these steps, we end up with 296,070 instances formed by two images and two captions, which we divided randomly into 80% for training and 20% for validation. Once again, we fine-tune a pretrained CLIP model, resulting on CLIP$_{\text{TROHN-IMG}}$ (training details in Appendix E).

Table 3: Results for contrastive models of the same size on SUGARCREPE and BIVLC, including our models (bottom rows). Bold for best, underline for second best. For BIVLC, we show the three main scores plus individual retrieval task scores: $I_{pos}2T$ refers to image-to-text retrieval with positive image, $I_{neg}2T$ for image-to-text retrieval with negative image, etc.

| Model | SUGARCREPE | BIVLC | | | BIVLC (finer metrics) | | | |
|---|---|---|---|---|---|---|---|---|
| | | I2T | T2I | Group | $I_{pos}2T$ | $I_{neg}2T$ | $T_{pos}2I$ | $T_{neg}2I$ |
| Random | 50.00 | 25.00 | 25.00 | 16.67 | 50.00 | 50.00 | 50.00 | 50.00 |
| CLIP | 76.56 | 75.83 | 52.40 | 49.06 | 84.32 | 89.50 | 69.21 | 82.82 |
| CLIP$_{COCO}$ | 84.66 | 82.75 | 63.89 | 60.96 | 89.06 | 91.75 | 72.79 | 90.86 |
| NEGCLIP | 85.64 | 80.74 | 61.95 | 58.75 | 89.53 | 89.53 | 70.10 | **91.34** |
| GNM | 81.83 | 81.32 | 60.86 | 57.96 | 88.00 | 91.24 | 80.33 | 80.33 |
| CLIP$_{TROHN-TEXT}$ | **93.40** | 78.18 | 62.19 | 57.48 | **93.25** | 83.87 | 71.05 | 90.59 |
| CLIP$_{TROHN-IMG}$ | 89.40 | **88.54** | **71.84** | **69.25** | 92.12 | **95.33** | **81.45** | 90.15 |

## 5.3 Results and analysis

Table 3 shows the results of our two proposed models, i.e. CLIP$_{TROHN-TEXT}$ and CLIP$_{TROHN-IMG}$ compared to comparable contrastive models of the same size, both in SUGARCREPE and BIVLC main metrics (leftmost four columns). As it can be observed, CLIP$_{TROHN-IMG}$ outperforms all the other contrastive models by large margins for the main three metrics of BIVLC. This shows the effectiveness of our training process. Interestingly, the improvement comes mainly from text-to-image retrieval, which greatly benefits from training with hard negative images. However, CLIP$_{TROHN-IMG}$ is still much better for image-to-text retrieval and its group score is well below human performance, which shows that further research is needed for achieving human-comparable performance for bidirectional VLC. As mentioned in Section 4, we were able to run VQAScore and the open version of CapPa on BiVLC (see Table 2). CLIP$_{TROHN-IMG}$ compares favorably to Open CapPa by a large margin, and it is comparable to VQAScore-XL, even though being orders of magnitude smaller (3B vs 151M parameters), but is clearly outperformed by the largest VQAScore. Regarding the two directions, CLIP$_{TROHN-IMG}$ is best on I2T but lags behind both VQAScore models in T2I. Given the large difference in size, it is not clear whether the model architecture of VQAScore is key for the high performance.

On the other hand, CLIP$_{TROHN-TEXT}$ is the best contrastive model for SUGARCREPE, outperforming NEGCLIP by 8 points and closing the gap with humans to 5 points. However, this strong performance does not reflect on BIVLC, where CLIP$_{TROHN-TEXT}$ is even worse than CLIP$_{COCO}$, NEGCLIP and GNM. This highlights our previous finding about the lack of correlation between SUGARCREPE and BIVLC results (Section 4).

To put it into the context of the state-of-the-art, the strongest models for SUGARCREPE are VQAScore-XXL [Lin et al., 2024] (93.72), CapPa [Tschannen et al., 2024] (92.88) and GPT-4V [Achiam et al., 2023] (92.19)[2]. CLIP$_{TROHN-TEXT}$ is on par with VQAScore-XXL and outperforms CapPa and GPT-4V, showing it is very strong for image-to-text retrieval, despite its significantly smaller size.

The lower performance of CLIP$_{TROHN-IMG}$ on SUGARCREPE can be attributed to TROHN-IMG containing 10 times less hard negative captions than TROHN-TEXT, and could improve with more instances.

**Why does training with hard negative images help?** We will now refer to the four finer metrics reported in the four rightmost columns in Table 3. The two multimodal systems trained with hard negative images, CLIP$_{TROHN-IMG}$ and GNM, obtain a difference of around 9 points for the $T_{pos}2I$ metric, i.e. the text-to-image retrieval accuracy for the positive caption. The contrastive training with hard negative images promotes learning features that maximize the distance between the positive caption and the negative image. This is not guaranteed when only hard negative texts are used for training. We show a conceptual diagram to illustrate this difference in Figure 4, which explains the benefits of training with hard negative images.

---

[2]Results obtained from `https://github.com/RAIVNLab/sugar-crepe/tree/main/gpt-4v-results`

Table 4: Results of multimodal models for BiVLC, divided into REPLACE, SWAP and ADD categories. Bold for best, underline for second best.

| Model | REPLACE | | | SWAP | | | ADD | | |
|---|---|---|---|---|---|---|---|---|---|
| | I2T | T2I | Group | I2T | T2I | Group | I2T | T2I | Group |
| Random | 25.00 | 25.00 | 16.67 | 25.00 | 25.00 | 16.67 | 25.00 | 25.00 | 16.67 |
| CLIP | 82.09 | 60.36 | 57.27 | 46.52 | 16.16 | 13.65 | 70.32 | 44.63 | 39.58 |
| CLIP$_{\text{COCO}}$ | 87.99 | 72.42 | 69.94 | 51.53 | 25.07 | 20.89 | 83.16 | 55.58 | 51.58 |
| NEGCLIP | 86.37 | 70.80 | 68.03 | 47.08 | 24.23 | 18.66 | 81.26 | 51.37 | 48.00 |
| GNM | 86.14 | 68.70 | 65.75 | 50.70 | 17.83 | 15.88 | 83.16 | 58.74 | 55.37 |
| CLIP$_{\text{TROHN-TEXT}}$ | 83.61 | 70.03 | 65.84 | 54.32 | 25.35 | 20.61 | 72.21 | 55.37 | 48.42 |
| CLIP$_{\text{TROHN-IMG}}$ | **91.62** | 79.18 | 76.61 | 62.40 | 32.03 | 27.86 | **94.74** | **69.47** | **68.00** |
| Open CapPa | 70.32 | 62.79 | 53.03 | 40.95 | 30.64 | 20.61 | 14.74 | 46.32 | 9.263 |
| VQAScore-XL | 85.95 | 84.18 | 78.28 | 65.18 | 58.77 | 48.75 | 77.05 | 56.63 | 50.74 |
| VQAScore-XXL | 89.85 | **88.33** | **83.85** | **72.14** | **66.30** | **56.82** | 80.42 | 65.47 | 58.74 |

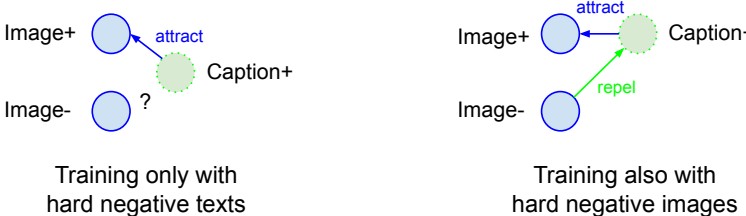

Figure 4: When we train only with hard negative texts, the distance of the positive caption (Caption+) and the negative image (Image-) may be even smaller than the distance of the positive caption to the positive image (Image+) (left). When we add hard negative images, we force to increase the distance between the positive caption and the negative image, while minimizing the distance between the positive caption and image (right).

**Which category is the most difficult?** Table 4 shows results of all multimodal models in BiVLC according to caption generation categories. We find that, similar to SUGARCREPE, the SWAP category is the hardest one for all models also for BiVLC. The group scores for all models are more than 20 points below the scores for the REPLACE category. But contrary to SUGARCREPE, we observe that the ADD category is also challenging for models in BiVLC. It is surprising to see the low performance of Open CapPA for that category, since its performance in SUGARCREPE is close to perfection. This highlights again the importance of language biases and evaluation protocols that mitigate the effects of those biases. In that sense, it is also interesting to see that CLIP$_{\text{TROHN-IMG}}$ is the best model for the ADD category, outperforming VQAScore-XXL by almost 10 points, despite its significantly smaller size.

**Why is CLIP$_{\text{TROHN-IMG}}$ still far from humans?** The steps followed to create BiVLC showed that current text-to-image systems such as SDXL-DPO do not always generate images faithfully. In fact, the manual filtering process used to produce the test dataset (see Section 3) allows to estimate the noise rate in TROHN-IMG to be around 61%, combining image generation failures and ambiguous instances. We think that this high noise rate in the training data prevents CLIP$_{\text{TROHN-IMG}}$ from learning better. This is reflected again in the T$_{\text{pos}}$2I metric, which is the weakest retrieval case (the other three retrieval scores are over 90). We can conclude that the generation of hard negative images is a promising direction, where further advances would be possible, like an automatic and scalable procedure to remove incorrect instances from TROHN-IMG, or, alternatively, new methods to produce cleaner training sets.

**Are our models just distinguishing between synthetic and natural?** Since the TROHN-IMG dataset contains synthetic and natural image-caption pairs, the good performance of CLIP$_{\text{TROHN-IMG}}$ on BiVLC could be attributed to its capacity to distinguish between synthetic and natural images and

Table 5: Results on BıVLC for synthetic vs. natural image and text detection-based systems. We show text and image detection accuracies, as well as the scores on the three main evaluation metrics.

| Model | Text detection acc | Img detection acc | I2T | T2I | Group |
|---|---|---|---|---|---|
| Random | 50.00 | 50.00 | 25.00 | 25.00 | 16.67 |
| CLIP$_{\text{Det}}$ | 57.00 | 100.00 | 66.69 | 19.64 | 19.64 |
| CLIP$_{\text{TROHN-IMG/Det}}$ | 61.34 | 100.00 | 75.04 | 26.42 | 26.42 |

captions, rather than to VLC skills. To analyse this, we develop two new systems which are trained to detect synthetic and natural images and captions. CLIP$_{\text{Det}}$ is composed by the original pretrained CLIP visual and text encoders, where a binary classification head is added to each encoder. Both binary classifiers are trained separately for synthetic image or caption detection reusing TROHN-IMG training images and texts (see Appendix E for details). CLIP$_{\text{TROHN-IMG/Det}}$, is built similarly, but using the visual and text encoders of our CLIP$_{\text{TROHN-IMG}}$ model.

The two detection columns in Table 5 show the performance of our methods when detecting synthetic text and images at BıVLC instances, with a perfect performance for synthetic image detection, i.e. both systems are able to detect which images in BıVLC are synthetic. The accuracy for text detection is lower, with CLIP$_{\text{TROHN-IMG/Det}}$ obtaining the highest accuracy with 61.34.

We can also evaluate the detectors on the BıVLC retrieval tasks: (i) for image-to-text retrieval, we first use the visual encoder to predict the type of image (natural or synthetic); if the detection is correct, the output of the text detector is used to see which of the captions has the highest probability to be of the same type as the image, and we select it as the matching caption; (ii) for text-to-image retrieval, we follow the same procedure, but using first the text detector over the given caption, and afterwards the visual detector for the two candidate images. The rightmost columns in Figure 5 show that BıVLC group score is very low for both detectors, underperforming clearly the multimodal models (cf. Table 3). This means that our models are actually learning much more than just distinguishing synthetic and natural images-captions. We already had another evidence of that in Table 3, where all models do equally well on I$_{\text{pos}}$2T and I$_{\text{neg}}$2T, i.e. they work equally well for both natural and synthetic images.

However, we also observe that the I2T score is significantly higher than random, scoring 75.04 in the best case. This result suggests that image-to-text retrieval is more sensitive to natural/synthetic bias, so we also checked the results of our detectors for SUGARCREPE. Surprisingly, we obtained an accuracy of 74.59 for CLIP$_{\text{Det}}$ and 77.65 for CLIP$_{\text{TROHN-IMG/Det}}$. Those results are actually in par with the original CLIP using similarity between the image and both captions (see Table 3). This is yet another reason to consider bidirectional datasets when evaluating compositionality.

## 6 Conclusions

We have presented BıVLC, a dataset for bidirectional vision-language compositionality. Our dataset offers the possibility of evaluating the compositional representation of multimodal models in both retrieval directions: image-to-text and text-to-image. We have shown that including text-to-image retrieval leads to novel findings, such as the unbalanced performance of multimodal models in the two retrieval directions, in contrast to humans. We also uncovered the low performance for BıVLC of the models which have been specifically trained for image-to-text retrieval. Furthermore, we have proposed a new way to train models with hard negative images, resulting on the best contrastive model for BıVLC so far. The gap with human performance shows that BıVLC can be used to measure further advances in modelling VLC.

As for future work, we would like to better analyse why text-to-image retrieval is harder than the other direction. We also plan to explore novel strategies to automatically generate and filter hard negative images for training, with the objective of improving the performance of multimodal systems in modelling compositionality in vision-language scenarios.

## Acknowledgements

This work is partially supported by the Ministry of Science and Innovation of the Spanish Government (AWARE project TED2021-131617B-I00, DeepKnowledge project PID2021-127777OB-C21), project funded by MCIN/AEI/10.13039/501100011033 and by FEDER, the Basque Government (IXA excellence research group IT1570-22), and the European Union under Horizon Europe (Project LUMINOUS, grant number 101135724).

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

# A  Limitations and societal impact

## A.1  Limitations

BIVLC offers captions only in English. It would be interesting to extend the dataset to other languages, as some recent works in vision-language models are already doing [Pouget et al., 2024, Chen et al., 2023, Bugliarello et al., 2022]. Moreover, we only trained contrastive models, due to their suitability for image-to-text and text-to-image retrieval tasks and their availability. In the future, generative multimodal models, which we evaluated but not fine-tuned, could also be explored. Indeed, our approach to fine-tune contrastive models with hard negative images also has its limitations: we evaluated the models in Winoground [Thrush et al., 2022] and we saw that improvements are modest. There are several hypotheses to explain those results and one of them points to the effect of using synthetic images. Deeper analyses are needed to elucidate the real effect of synthetic images to train multimodal models. Finally, as we rely on SUGARCREPE hard negative captions, we also use the same categories. Adding more diversity by extending BIVLC to other categories could be beneficial.

## A.2  Societal impacts

Vision-language models such as CLIP are becoming popular models for many applications, but previous research has probed and analyzed their limitations [Yuksekgonul et al., 2022, Hsieh et al., 2024]. We contribute with our research by delving into a new point of view: the importance of measuring bidirectional compositionality. We hope that our benchmark will lead to a better assessment of the compositional understanding of vision-language models and may thus lead to their improvement. Furthermore, we expect BIVLC to become one of the main benchmarks used to improve the compositional capabilities of vision-language models, as it enables deeper analysis of their behaviour and it is more challenging than previous VLC tasks.

# B  BIVLC dataset information

We host BIVLC at HuggingFace[3]. The Croissant metadata record, which contains dataset metadata, is available in the dataset repository[4]. The DOI for BIVLC dataset is 10.57967/hf/2391, can be found in the dataset repository[5]. We provide a summary below.

**Dataset documentation**   BIVLC is a benchmark for Bidirectional Vision-Language Compositionality evaluation. Each instance consists of two images and two captions. Using each of the images and captions as a base, a model is asked to select the pair that correctly represents the base versus the hard negative distractor with minor compositional changes. Thus, we can measure image-to-text and text-to-image retrieval with hard negative pairs. To obtain good results on the dataset, it is necessary that the model performs well in both directions for the same instance. Each instance of the dataset consists of six fields:

- image: COCO 2017 validation image.
- caption: COCO 2017 validation text describing the COCO image.
- negative_caption: Negative caption generated from the COCO text description by SUGAR-CREPE [Hsieh et al., 2024].
- negative_image: Negative image generated from the negative caption by us for BIVLC.
- type: Category of the negative instances: REPLACE, SWAP or ADD.
- subtype: Subcategory of the negative instances: OBJECT, ATTRIBUTE or RELATION.

Example of a BIVLC instance after load_dataset("imirandam/BiVLC", split = "test") (Figure 5):

**Maintenance plan**   We are committed to maintaining the dataset to resolve any technical issues. We actively track issues in the HuggingFace or GitHub repositories provided in `https://imirandam.github.io/BiVLC_project_page`.

---

[3]`https://huggingface.co/datasets/imirandam/BiVLC`
[4]`https://huggingface.co/api/datasets/imirandam/BiVLC/croissant`
[5]`https://huggingface.co/datasets/imirandam/BiVLC?doi=true`

```
{
    'image': <PIL.JpegImagePlugin.JpegImageFile image mode=RGB size=500x332 at 0x7F9BFC0C5430>,
    'caption': 'A man throwing a ball while smiling and on a field.',
    'negative_caption': 'A man throwing a ball while a child is smiling on a field.',
    'negative_image': <PIL.JpegImagePlugin.JpegImageFile image mode=RGB size=512x512 at 0x7F9BE45571C0>,
    'type': 'add',
    'subtype': 'obj',
}
```

Figure 5: Example of a BIVLC instance after loading the dataset.

**Licensing**   We license our work using the MIT License [6].

**Author statement**   We, the authors, assume full responsibility in case of violation of rights.

## C   TROHN-TEXT dataset details

The detailed statistics for the TROHN-TEXT dataset can be found in Table 6. The number of instances per category and subcategory are provided.

Table 6: Statistics for the TROHN-TEXT dataset, divided into the different categories and subcategories. Each instance is composed by one image and two captions.

| | REPLACE | | | SWAP | | ADD | | TOTAL |
|---|---|---|---|---|---|---|---|---|
| | OBJ | ATT | REL | OBJ | ATT | OBJ | ATT | |
| # instances | 570,325 | 571,609 | 576,101 | 333,705 | 429,163 | 584,077 | 587,866 | 3,652,846 |

## D   Detailed evaluation metrics

The **I2T** score measures the performance for image-to-text retrieval. For each instance in our dataset, we actually have two image-to-text retrieval examples. To obtain a perfect I2T score, the correct captions for both images have to be selected. Thus, assuming $C_0, C_1$ refer to positive and negative caption respectively, $I_0, I_1$ to positive and negative image, and we use $s(C_i, I_i)$ as the similarity function for a caption and an image, I2T score $I2T(C_0, I_0, C_1, I_1)$ is defined in Equation 1:

$$I2T(C_0, I_0, C_1, I_1) = \begin{cases} 1 & \text{if } s(C_0, I_0) > s(C_1, I_0) \\ & \text{and } s(C_1, I_1) > s(C_0, I_1) \\ 0 & \text{otherwise} \end{cases} \quad (1)$$

The **T2I** score $T2I(C_0, I_0, C_1, I_1)$ is similarly defined for text-to-image retrieval (Equation 2):

$$T2I(C_0, I_0, C_1, I_1) = \begin{cases} 1 & \text{if } s(C_0, I_0) > s(C_0, I_1) \\ & \text{and } s(C_1, I_1) > s(C_1, I_0) \\ 0 & \text{otherwise} \end{cases} \quad (2)$$

Finally, the **Group** score $G(C_0, I_0, C_1, I_1)$ is the main metric, since it combines the performance for image-to-text and text-to-image retrieval. To obtain a perfect group score for a given instance, both images have to be matched with the suitable captions and both captions with the suitable images. The group score is defined in Equation 3:

$$G(C_0, I_0, C_1, I_1) = \begin{cases} 1 & \text{if } I2T(C_0, I_0, C_1, I_1) \\ & \text{and } T2I(C_0, I_0, C_1, I_1) \\ 0 & \text{otherwise} \end{cases} \quad (3)$$

---

[6]`https://github.com/IMirandaM/BiVLC/blob/main/LICENSE`

# E   Implementation details

This appendix contains all the information related to the implementation of the experiments. All the source code with instructions can be found at `https://github.com/IMirandaM/BiVLC`.

## E.1   Source datasets

We obtain all source datasets directly from the original sources published by the authors. To the best of our knowledge, all data sources we use are open to non-commercial use, do not contain personally identifiable information and do not contain offensive content.

- **COCO** [Lin et al., 2014]: We obtain COCO 2017 from the official project website[7] under a Creative Commons Attribution 4.0 License[8].
- **SUGARCREPE** [Hsieh et al., 2024]: We obtain the negative captions from the official GitHub project[9] under the MIT license[10].

## E.2   Train and Validation datasets

Three different variants, $CLIP_{COCO}$, $CLIP_{TROHN-TEXT}$ and $CLIP_{TROHN-IMG}$, have been fine-tuned. For that we used 3 different training and validation datasets:

- $CLIP_{COCO}$ is fine-tuned using COCO 2017 train which contains 591,753 captions and 118,287 images, i.e., 591,753 instances formed by an image and a caption.
- $CLIP_{TROHN-TEXT}$ is fine-tuned with the TROHN-TEXT dataset consisting of 3,652,846 instances formed by one image and two captions. See Section 5.1 for more details.
- $CLIP_{TROHN-IMG}$ is fine-tuned with the TROHN-IMG dataset consisting of 296,070 instances formed by two images and two captions, i.e. 592,140 pairs, an amount similar to that of the COCO 2017 train. See Section 5.2 for more details.

All three datasets are randomly divided into 80% for training and 20% for validation. The TROHN-TEXT and the TROHN-IMG datasets are in HuggingFace repositories[11][12], with all the necessary information for its use.

## E.3   Software information

**Models**   We detail the sources of the pretrained and fine-tuned models we used.

- **OPENCHAT-3.5-0106** We obtain the model released by [Wang et al., 2023a] [13].
- **CoLA** We obtain RoBERTa base model [Liu et al., 2019] fine-tuned in CoLA released by [Morris et al., 2020] [14].
- **VERA** We obtain pretrained Vera model released by [Liu et al., 2023][15].
- **Pretrained CLIP from OpenCLIP** We obtain the pretrained baseline VIT-B-32 OpenAI's CLIP model [Radford et al., 2021] from OpenCLIP [Cherti et al., 2022][16].
- Fine-tuned CLIP models:
    1. **NEGCLIP** We obtain fine-tuned CLIP model released by [Yuksekgonul et al., 2022][17].

---

[7]`https://cocodataset.org/#download`
[8]`https://cocodataset.org/#termsofuse`
[9]`https://github.com/RAIVNLab/sugar-crepe/tree/main/data`
[10]`https://github.com/RAIVNLab/sugar-crepe/blob/main/LICENSE`
[11]`https://huggingface.co/datasets/imirandam/TROHN-Text`
[12]`https://huggingface.co/datasets/imirandam/TROHN-Img`
[13]`https://huggingface.co/openchat/openchat-3.5-0106`
[14]`https://huggingface.co/textattack/roberta-base-CoLA`
[15]`https://huggingface.co/liujch1998/vera`
[16]`https://github.com/mlfoundations/open_clip`
[17]`https://github.com/mertyg/vision-language-models-are-bows`

2. **GNM** We obtain fine-tuned CLIP model released by [Sahin et al., 2024][18].

- **VQAScore** We obtain model released by [Lin et al., 2024][19].

- **Open CapPa** We obtain the open source CapPa model from the official reposiroty [20].

- **CapPa** SUGARCREPE results obtained from [Tschannen et al., 2024].

- **GPT-4V** results are taken from the SUGARCREPE web page [21].

**Fine-tuning hyperparameters**   We fine-tuned three CLIP models: $CLIP_{COCO}$, $CLIP_{TROHN-TEXT}$ and $CLIP_{TROHN-IMG}$. We also trained two detectors, $CLIP_{Det}$ and $CLIP_{TROHN-IMG/Det}$, based on synthetic-natural image-text detection.

We base CLIP fine-tuning on OpenCLIP [Cherti et al., 2022]. Detailed hyperparameters:

- Learning rate: 1e-6.

- Scheduler: Cosine scheduler with 50 warmup steps.

- Optimizer: AdamW optimizer with beta1 = 0.9, beta2 = 0.98, eps = 1e-6 and weight decay = 0.1.

- Loss function: InfoNCE Loss. In the case of $CLIP_{TROHN-TEXT}$ the loss is modified to add only negative captions following the idea proposed in NEGCLIP [Yuksekgonul et al., 2022].

- Batch size: We define a batch size of 400 (400 images x 400 captions) for $CLIP_{COCO}$. For $CLIP_{TROHN-TEXT}$ and $CLIP_{TROHN-IMG}$ we define a batch size of 200, and then we add negatives. In the case of $CLIP_{TROHN-TEXT}$, as it has not hard negative images, it results in 200 images x 400 captions (positive + hard negatives). For $CLIP_{TROHN-IMG}$ the batch consists of 200 positive pairs and 200 negative pairs, resulting in 400 images x 400 captions.

- Epochs: We fine-tune all models over 10 epochs and we used validation accuracy as the model selection criterion, i.e. we selected the model with the highest accuracy on the corresponding validation set.

We also trained $CLIP_{Det}$ and $CLIP_{TROHN-IMG/Det}$ detectors for binary classification by keeping the encoders frozen and adding a sigmoid neuron over the CLS embedding for the image encoder and over the EOT embedding for the text encoder. Detailed hyperparameters:

- Learning rate: 1e-6.

- Optimizer: Adam optimizer with beta1 = 0.9, beta2 = 0.999, eps = 1e-08 and without weight decay.

- Loss function: Binary cross-entropy loss (BCELoss).

- Batch size: We define a batch size of 400.

- Epochs: We trained the text detector over 10 epochs and the image detectors over 1 epoch. We used validation accuracy as the model selection criterion, i.e. we selected the model with the highest accuracy in the corresponding validation set.

**Evaluation**   For contrastive models, we base our evaluation on OpenCLIP [Cherti et al., 2022]. We follow all the default hyperparameters used to evaluate models, making sure that when loading the checkpoints we are using QuickGELU as in the base pretrained model[22]. For the generative models, we have used the official GitHub repositories of each model and followed the instructions and relied on the evaluation codes provided by their authors.

---

[18]https://github.com/ugorsahin/Generative-Negative-Mining

[19]https://github.com/linzhiqiu/t2v_metrics

[20]https://github.com/borisdayma/clip-jax

[21]https://github.com/RAIVNLab/sugar-crepe/tree/main/gpt-4v-results

[22]Evaluation bug when using GELU vs QuickGELU https://github.com/RAIVNLab/sugar-crepe/issues/7

### E.4 Hardware information

All the experiments of this research have been performed on internal clusters of HiTZ Zentroa. Experiments are run on two servers. For the main experiments, we work in a Supermicro Superserver AS-4124GO-NART with 2 x AMD EPYC 7513, 1024 GB RAM and 8x A100-SXM4-80GB GPUs. For smaller runs, we work in a Dell Poweredge R740 with 2x Intel Xeon Gold 6226R, 256 GB RAM and 2x A30 GPUs. We estimate that for the generation of the datasets (BiVLC, TROHN-TEXT and TROHN-IMG), and the training of the different models, around 387 hours of execution have been needed. Detailed information for specific runs:

**CLIP fine-tuning** For fine-tuning CLIP models we use one NVIDIA A100-SXM4-80GB GPU. The execution time varies depending on the variant: for $CLIP_{TROHN-TEXT}$, due to the amount of data, it takes about 6 days, in the case of $CLIP_{COCO}$ and $CLIP_{TROHN-IMG}$ as the datasets are smaller about 18 hours each.

**BiVLC** It was necessary to create 30,048 images in total. For that purpose, we used 4 NVIDIA A100-SXM4-80GB GPUs with an approximate execution time of 11 hours.

**TROHN-TEXT dataset** We used one NVIDIA A100-SXM4-80GB GPU. For the generation of each of the seven subtypes proposed in SUGARCREPE it takes approximately 16 hours, estimating a total of 112 hours.

**TROHN-IMG dataset** It was necessary to generate 296,070 images. Thus we used 6 NVIDIA A100-SXM4-80GB GPUs with an execution time of approximately 72 hours.

**Detectors** We used one NVIDIA A30 GPU. The sigmoid neuron of the text detector is trained for 10 epochs taking approximately 1.5 hours, and for the image detector it is trained for one epoch taking approximately 3 hours.

**Evaluation** All evaluations have been performed on one NVIDIA A100-SXM4-80GB GPU.

## F  SUGARCREPE templates

For the generation of hard negative captions, for the training and validation splits, we used the templates proposed by SUGARCREPE. Each subcategory defined in SUGARCREPE has a template that can be found in [Hsieh et al., 2024].

## G  BiVLC instance examples

Examples of each category and their corresponding subcategories are presented in Figure 6. If you wish to see more examples in a clearer way, please access the viewer mode of the BiVLC repository[23].

---

[23]https://huggingface.co/datasets/imirandam/BiVLC/viewer

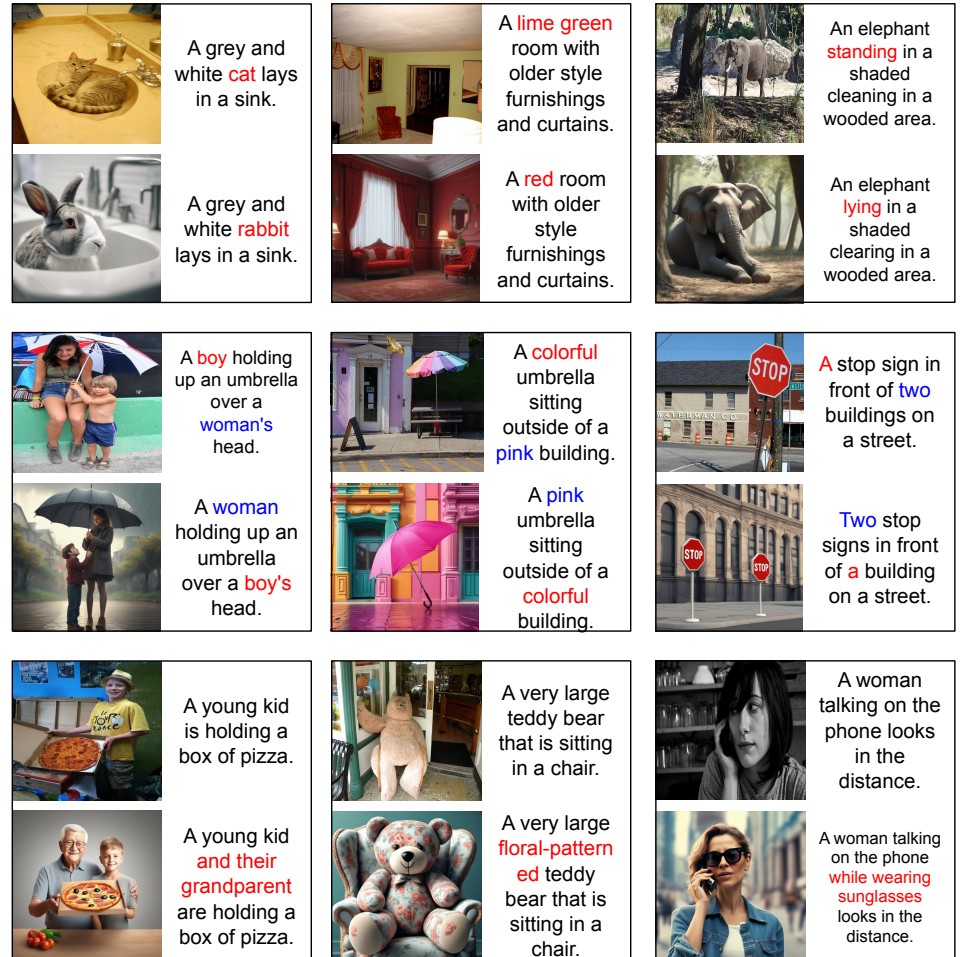

Figure 6: BIVLC instance examples. From top to bottom three examples of the categories REPLACE (first row), SWAP (second row) and ADD (third row).

# H Crowdsourcing

We have carried out crowdsourcing on the Prolific platform[24]. The following sections will detail the instructions and payments.

## H.1 Instructions for the different stages

We have used crowdsourcing in three different stages, here are the instructions for each stage.

**Choose the best generated image**    The instructions for this stage are detailed below (see Step 3 in Section 3). An example of a question as seen by the annotators is depicted in Figure 7.

```
"Each instance will consist of a text description and 4 images (A,B,C,D).
The aim is to choose the image that represents the information provided in
the description, and discard the rest. It might be that more than one image
is acceptable or none is acceptable. We only need you to select one, if
available, or none.
```

---

[24]https://www.prolific.com/

```
Instructions:

Select one image that adequately represents the description. In case none
of the images adequately represent the description, select the answer
"None".

Important:

1) All information presented in the text description must appear in the
image.

2) Aesthetic defects are accepted, such as deformed faces, arms, hands
etc."
```

**Filter ambiguous instances**    The instructions for this stage are detailed below (see Step 4 in Section 3). An example of a question as seen by the annotators is depicted in Figure 8.

```
"Each instance will consist of a description and two images. The objective
is to choose the image that best represents the description or, in case
both images represent the description equally well, choose "Both".

Instructions:

Choose the image that best matches the description, in the case that both
images match the description equally well choose "Both".

Important:

1) Aesthetic defects are accepted, such as deformed faces, arms, hands
etc."
```

**Human Baseline**    The instructions for this stage are divided into two: image-to-text retrieval (I2T) and text-to-image retrieval (T2I). Both are detailed below, and examples of question as seen by the annotators are depicted in Figures 9 and 10. The obtained human performance for each task can be seen in Table 2.

```
"Each survey consists of 100 questions/instances. We split the instances
into 2 parts (50/50), in which the task changes.

The first part of the instances consists of an image and two descriptions.
The goal is to choose the description that BEST represents the image.

1) Instructions: Choose the description that BEST represents the image.
2) Important: Pay close attention to word order.

The second part of the instances consists of a description and two images.
The objective is to choose the image that BEST represents the description.

1) Instructions: Choose the image that BEST matches the description.
2) Important: Aesthetic defects are accepted, such as deformed faces, arms,
hands etc."
```

## H.2   Hourly wage paid to participants and total amount spent

The hourly wage paid to all participants has been US$12.0, the recommended rate in the Prolific platform. The total amount spent on crowdsourcing is US$1,331.61, which includes the payment to participants plus service commissions.

### H.3 Inter-tagger agreement

During "Step 3 - Ask to human annotators to choose the best generated image" (see Section 3.1) we used 10 annotators divided into pairs to score a total of 500 annotations and obtain inter-tagger agreement. To do this, we calculated Cohen's Kappa score between the responses of each pair who took the same survey (Table 7). The score is calculated based on whether they chose one of the images or none. The mean score obtained among the 5 groups of annotators is 0.49, which was interpreted as moderate agreement.

Table 7: Cohen's Kappa score for each of the pairs used to calculate inter-tagger agreement and the mean score of the 5 pairs.

| Pair | Kappa score | Mean |
|------|-------------|------|
| 1 | 0.66 | |
| 2 | 0.43 | |
| 3 | 0.56 | **0.49** |
| 4 | 0.43 | |
| 5 | 0.38 | |

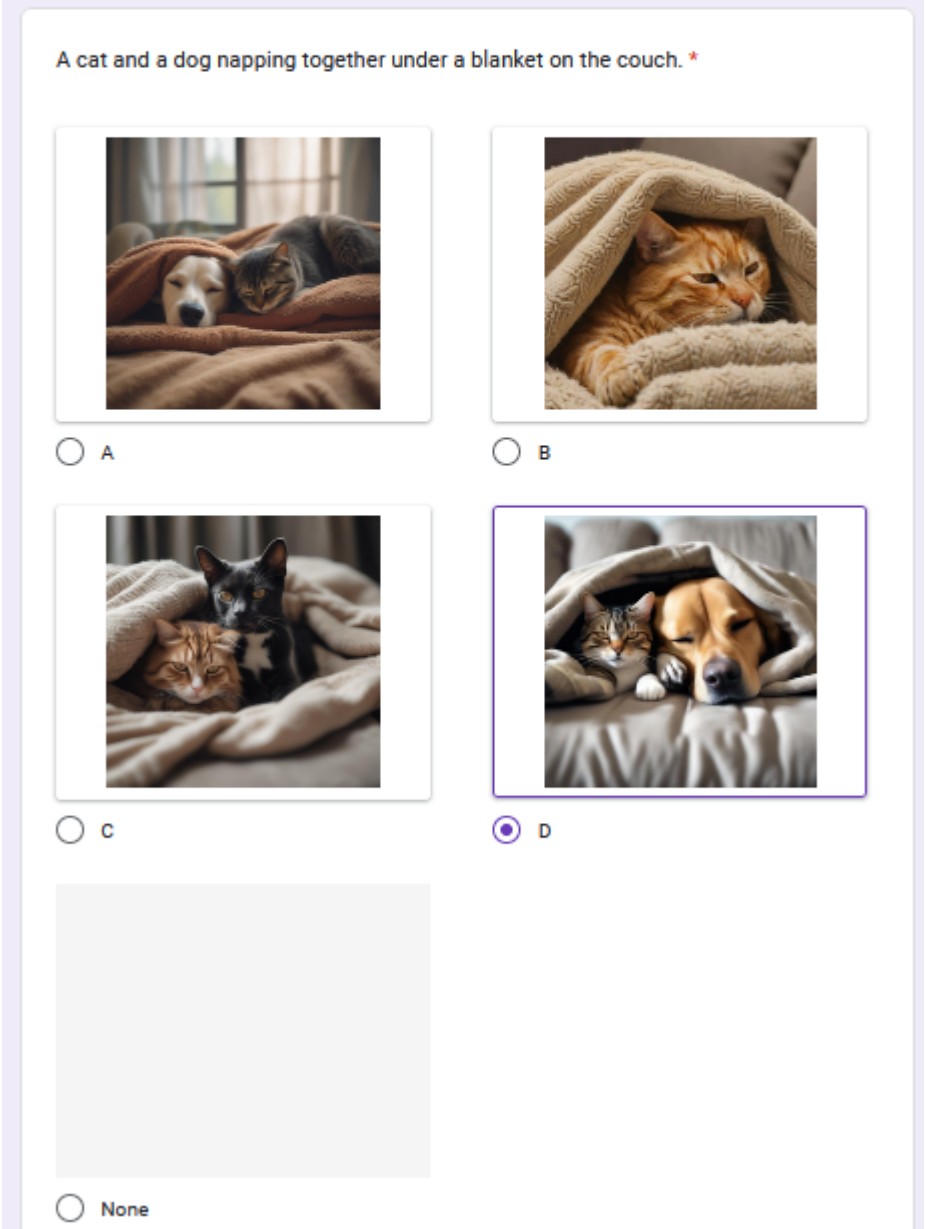

Each instance will consist of a text description and 4 images (A, B, C, D). The aim is to choose the image that represents the information provided in the description, and discard the rest. It might be that more than one image is acceptable or none is acceptable. We only need you to select one, if available, or none.

Instructions:
Select one image that adequately represents the description. In case none of the images adequately represent the description, select the answer "None".

Important:
1) All information presented in the text description must appear in the image.
2) Aesthetic defects are accepted, such as deformed faces, arms, hands etc.

A cat and a dog napping together under a blanket on the couch. *

○ A

○ B

○ C

◉ D

○ None

Figure 7: Example provided in one of the surveys conducted to select the best negative image.

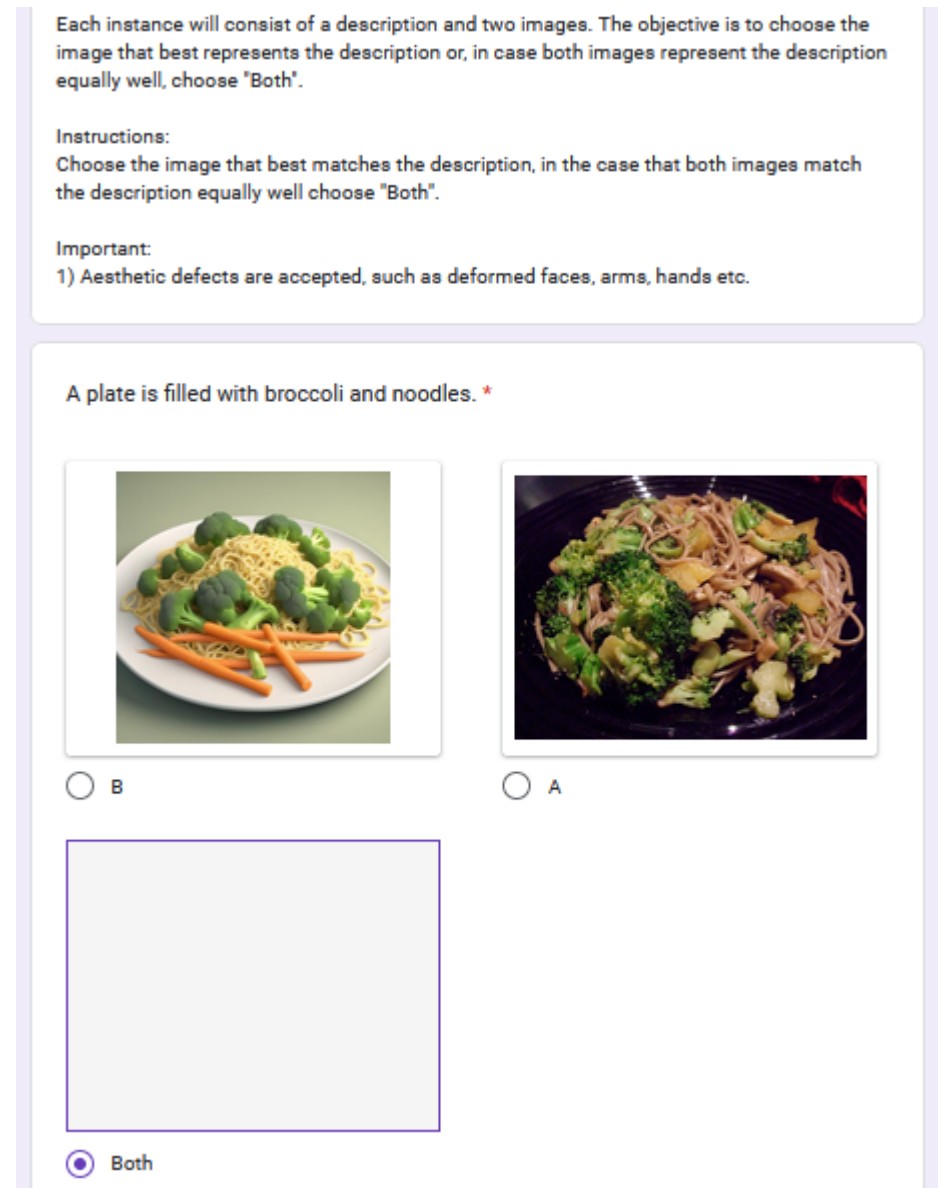

Each instance will consist of a description and two images. The objective is to choose the image that best represents the description or, in case both images represent the description equally well, choose "Both".

Instructions:
Choose the image that best matches the description, in the case that both images match the description equally well choose "Both".

Important:
1) Aesthetic defects are accepted, such as deformed faces, arms, hands etc.

A plate is filled with broccoli and noodles. *

○ B

○ A

◉ Both

Figure 8: Example provided in one of the surveys conducted to disambiguate instances based on the original caption.

Figure 9: Example provided in one of the surveys conducted to obtain the human baseline in the I2T direction.

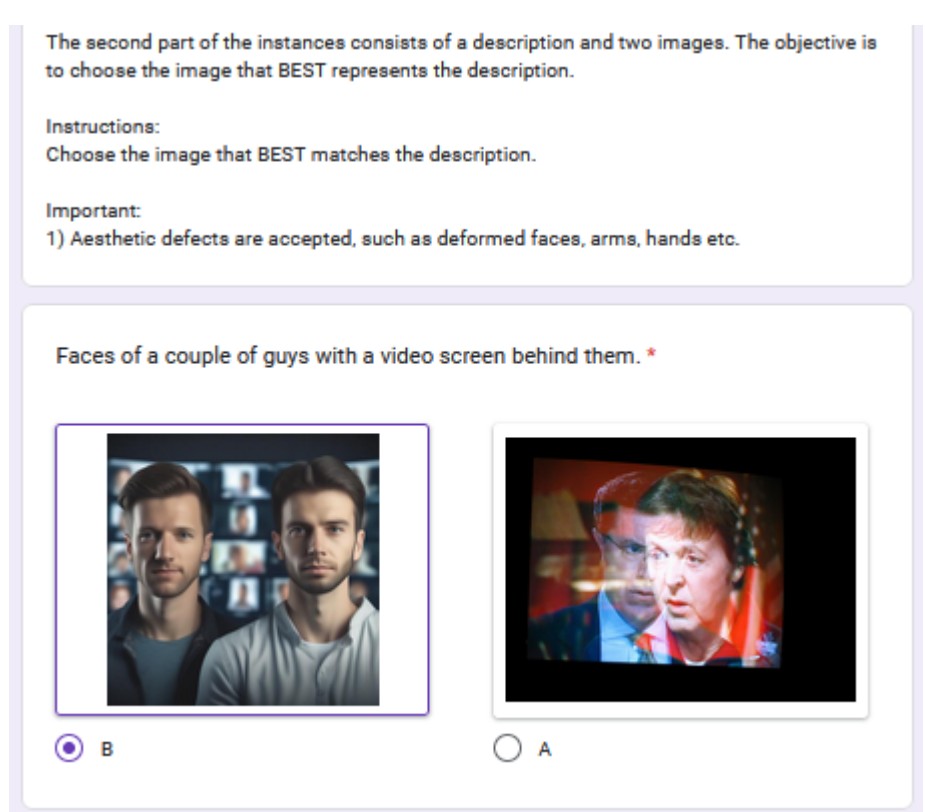

Figure 10: Example provided in one of the surveys conducted to obtain the human baseline in the T2I direction.

