# OpenReview forum: "BiVLC: Extending Vision-Language Compositionality Evaluation with Text-to-Image Retrieval"
_NeurIPS.cc/2024/Datasets_and_Benchmarks_Track — NeurIPS 2024 Track Datasets and Benchmarks Poster_

### Official Review · Reviewer_6Z5K · 2024-06-14
**The dataset is a nice contribution to the VLM community**

**Rating:** 7
**Confidence:** 5
**Clarity:** Yes, only a few typos that I mentione…

**Review:**

The proposed dataset and benchmark are of high quality and the paired (image, caption) evaluation protocols (adopted from Winoground) avoid language biases in previous benchmarks like SugarCrepe and ARO. The collection pipeline is efficient and scalable. The human annotation pipeline is well documented. There are a few test samples that should be discarded (I include them below). In terms of presentation, the paper is easy to follow and nicely written. It can however benefit from more careful proofreading; I include a few writing suggestions in the Improvement section.

**Strengths:**

Overall, I appreciate the authors’ efforts in collecting a retrieval benchmark in the same format as Winoground. Using text-to-image models to scale up benchmarks is a simple and natural idea. I also appreciate the authors sharing the dataset on HuggingFace with a nice dataset viewer. I randomly went through around 20 samples and found the dataset of reasonably high quality. However, there are still a few low-quality images slipped through the dataset (discussed in the Improvement section). The paper also provides a valuable discussion on how simply detecting synthetic vs. real images is not enough to solve this benchmark. The experimental results in this paper are convincing; however, I don’t know if the authors plan to release the training dataset.

**Additional Feedback:**

My current rating is not final. I hope the authors may engage with the reviewers during the discussion period to further improve the quality of this paper.

**Correctness:**

I believe the evaluation protocols adopted from Winoground is strong and more robust than previous benchmarks. I would encourage the authors to revise some statements and include more thought-provoking discussion to strengthen the paper.

**Documentation:**

Yes, the authors release a hugging face page to make the benchmark visible to all.

**Ethics:**

No.

**Limitations:**

I would like to know if the proposed training method and dataset improve on Winoground, e.g., in Table 3. It is possible that the method does not produce good results on Winoground since it relies heavily on synthetic data, whereas Winoground is mostly natural images. This is **okay**, as long as authors acknowledge this in the limitation.

**Opportunities For Improvement:**

**The paper can be made stronger by including a few important references:**
1. Winoground is not the only benchmark that supports “bidirectional” retrieval. There are two other published benchmarks: EqBen [1] and CoLA [2], although these two benchmarks are of lower quality due to visually difficult images (e.g., out-of-focus) or insufficient human verification. In addition, classic “large-scale” retrieval benchmarks such as COCO-retrieval and Flickr-retrieval also support bidirectional retrieval. This is not to say the proposed benchmark is not novel; I believe the “semi-automated” process to filter out low-quality samples is very crucial. The paper can perhaps be made stronger by conducting a small-scale human evaluation on Winoground/EqBen/CoLA to see if BiVLC has the highest human quality.

2. The paper primarily reports using **discriminative VLMs** like CLIP and NegCLIP. VQAScore [3] and VisualGPTScore [4] show stronger retrieval performance on ARO/ SugarCrepe / Winoground / EqBen using **generative VLMs** like BLIP2, LLaVA-1.5, and CLIP-FlanT5. In fact, [3] performs an in-depth analysis of the language biases in benchmarks like SugarCrepe and ARO (which allow these benchmarks to be solved by image-blind solutions). These prior work can serve as an even stronger motivation for the collection of BiVLC. [4] also releases the code for evaluating **open-source** generative VLMs on retrieval benchmarks like Winoground, unlike Cappa [5] which depends on proprietary models.

3. There are previous works that rely on synthetic images for training and evaluation such as SyViC [6]. Please refer to [3] for more training baselines that leverage data generated by foundation models or simulation engines.


**Feedback for writing:**

Some statements need to be revised:
1. Line 59: In addition to SugarCrepe, Cappa [5] and VisualGPTScore [4] also discuss the language biases in concurrent benchmarks.
2. Line 71-72: Winoground is not the only dataset for bidirectional retrieval. The related work may introduce concurrent benchmarks like EqBen [1], Cola [2].
3. Line 252: I believe VisualGPTScore based on BLIP [4] is still the SOTA on SugarCrepe.

Some minor typos:
1. Line 52: sistematicity -> systematically
2. Line 79: "increasing the size of Winoground over more than 7 times” -> this statement is misleading, I believe you meant to say that your benchmark is “larger than Winoground”.

Another suggestion: I will suggest including Random Chance performance in all tables to ease understanding.

**Feedback on dataset**:

Using generative models to construct benchmarks inevitably leads to low-quality images. Although the authors carefully filter them by hiring human annotators, I still notice a few low-quality samples slip through. Here are a few of them that should be discarded or re-generated:
1. A man and a child with tennis rackets walk from the net of a tennis court. (The generated image has the man and child walking in parallel to the tennis court; also the tennis rackets are not well rendered).
2. Room with stone floors and a wood fire place. (The generated negative image seems to have a stone fire place just like the positive image.)
3. On top of a red desk, a pair of scissors. (One of the scissors in the negative image is not well rendered; the pair of scissors in the positive is only partially shown)

Though these low-quality samples are rare, I will encourage the authors to do one more check and perhaps remove more low-quality images. After all, quality matters more than quantity for a testing benchmark.

**References:**

[1] Equivariant similarity for vision-language foundation models. Wang et al. ICCV 2023.

[2] cola: A Benchmark for Compositional Text-to-image Retrieval. Ray et al. NeurIPS 2023.

[3] Evaluating Text-to-Visual Generation with Image-to-Text Generation. Lin et al. ECCV 2024.

[4] Revisiting the Role of Language Priors in Vision-Language Models. Lin et al. ICML 2024.

[5] Image captioners are scalable vision learners too. Tschannen et al. NeurIPS 2024.

[6] Going Beyond Nouns With Vision & Language Models Using Synthetic Data. Cascante-Bonilla et al. ICCV 2023.

**Relation To Prior Work:**

Yes, but a few missing citations should be included.

**Summary And Contributions:**

The paper proposes a new image-text retrieval benchmark termed BiVLC that evaluates both image-to-text and text-to-image retrieval tasks. The benchmark is structured in the same format as Winoground (two images paired with two captions) and adopts the same evaluation protocols. The dataset is built upon SugarCrepe’s real images, captions, and synthetic negative captions. The authors additionally generate a synthetic negative image using a text-to-image generative model (SDXL-DPO) to pair with the negative caption. The collection is semi-automated because the authors hire annotators to filter out low-quality synthetic images. Experimental results suggest that this new benchmark is harder than SugarCrepe (albeit easier than Winoground), and current discriminative VLMs like CLIP and NegCLIP still lag behind human performance. The authors also propose to finetune a CLIP on the dataset constructed in a similar manner (though without human filtering), and achieve better performance on both SugarCrepe and BiVLC.

---

> ### Author Response · Authors · 2024-08-16
> **Response to Reviewer 6Z5K**
>
> **We thank the reviewer for the thorough review. The references, ideas and improvements proposed are very useful to strengthen the paper. We explain here how we will tackle them:**
>
> ### **Strengths:**
> **“The experimental results in this paper are convincing; however, I don’t know if the authors plan to release the training dataset.”**
>
> We do mention in the appendices that the training datasets are released, including the specific urls. The datasets can also be retrieved from the project main webpage and the github page, which we do mention in the paper. In order to be more explicit, we will mention in the main body of the paper that both training datasets are released and the urls to reach them.
>
> ### **Opportunities For Improvement:**
>
> **1. “Winoground is not the only benchmark that supports “bidirectional” retrieval.”**
>
> Thanks for the references. We will properly mention EqBEN and CoLA in the paper (see also the response in “Feedback for writing”). Note that COCO-retrieval and Flick-retrieval are not valuable for compositionality studies and they don’t provide, in general, hard negative image-captions. Regarding the small-scale human evaluation proposed by the reviewer, we think it is a very interesting idea that will empirically show the weaknesses of Winoground, EqBen and CoLA. We will collect pairwise preference data for texts and images coming from the different datasets and add the results to the final version.
>
> **2. “The paper primarily reports using discriminative VLMs like CLIP and NegCLIP.”**
>
> Thanks again for the valuable references. Following your recommendations, we evaluated two generative VLMs in our dataset: 1) OpenCapPa (https://wandb.ai/craiyon/cappa-jax/reports/CapPa-Training-vision-models-as-captioners---Vmlldzo4NDUyNDUz), which is a recent open implementation of CapPa almost matching its performance in SugarCrepe, and 2) VQAScore, for which we evaluated CLIP-FlanT5-XXL and CLIP-FlanT5-XL. Here we have the results (we also add CLIP-TROHN-Img as reference):
>
> | **Model**       | **SugarCrepe** |         | **BiVLC** |           |
> |-----------------|:--------------:|:-------:|:---------:|:---------:|
> |                 |                | **I2T** |  **T2I**  | **Group** |
> | Human           |      98.93     |  90.40  |   93.00   |   86.80   |
> | Random          |      50.00     |  25.00  |   25.00   |   16.67   |
> | OpenCapPa       |      90.46     |  57.62  |   55.71   |   41.87   |
> | CLIP-FlanT5-XL |      90.86     |  81.79  |   76.75   |   70.24   |
> | CLIP-FlanT5-XXL |      93.72     |  86.26  |   82.13   |   76.61   |
> | CLIP-TROHN-Img  |      89.40     |  88.54  |   71.84   |   69.25   |
>
> The results are very interesting, but first of all, we would like to highlight that the generative models are much larger than the contrastive models we explore (we will add a new column to Table 2 to reflect this) and use the larger ViT-L/14 vision encoder, instead of the ViT-B/32 we use, making head-to-head comparison troublesome.
>
> OpenCapPa has a poor performance in BiVLC, despite being very strong in SugarCrepe. This can be explained by the language bias of its LLM decoder, which can be beneficial in SugarCrepe, but is strongly penalized in our bidirectional setting, as already shown in [4].
>
> On the other hand, the largest VQAScore model obtains very strong results in SugarCrepe and BiVLC (+7 points better than CLIP-TROHN-Img), but still far from human performance (showing the value of our dataset). Even though it narrows the gap between I2T and T2I, we can still observe that T2I is more difficult for this VLM too. When we use the smaller CLIP-FlanT5-XL (decreasing from 11B to 3B), the results are slightly better than our CLIP-TROHN-Img in BiVLC. Overall, we can say that the results of the generative VLMs do not change the conclusions of the paper and show the value of our dataset to evaluate VL compositionality. We can also say that training with hard negative images is an interesting alternative, since it allows us to boost the performance of a small CLIP model to match a significantly larger model (CLIP-FlanT5-XL).
>
> Given the results of those experiments, we plan to do the following changes in the paper:
> * Add the results of OpenCapPa and VQAScore in Table 2.
> * Update the paragraph starting on line 146 to include the new generative models (OpenCapPa and VQAScore with its variants).
> * Update the text for Findings 1, 2 and 3. The main conclusions are still the same, but some details change.
> * Update the text in the abstract and introduction referring to CLIP-TROHN-Img, since it is not the sota model in BiVLC anymore.

---

> > ### Author Response · Authors · 2024-08-16
> > **Response to Reviewer 6Z5K (2)**
> >
> > **3. “There are previous works that rely on synthetic images for training and evaluation such as SyViC [6].”**
> >
> > We will include them in the paper. Concretely, we will rewrite the beginning of Section 5.2, where we introduce the TROHN-Img dataset (line 210):
> > “GNM [Sahin et al., 2024] used automatically generated images as hard negative examples, but to generate them, GNM uses image editing techniques and is limited to the Replace category. SyViC [ref] also explores training with synthetic images, but their images are generated with a graphic engine and require techniques such as style transfer to make them more natural.”
> >
> > ### **Feedback for writing:**
> >
> > **1. “Line 59: In addition to SugarCrepe, Cappa [5] and VisualGPTScore [4] also discuss the language biases in concurrent benchmarks.”**
> >
> > We will include the mentioned references.
> >
> > “Due to the use of heuristic rules for hard negative text generation, [10] found that very high accuracies can be obtained for those two datasets, even without using the images at all. Taking advantage of the biases introduced by the rule-based hard negative texts, such as nonsensical phrases or grammatically incorrect texts, purely textual models outperform the best multimodal models. Similar language biases have also been studied for concurrent benchmarks by [4, 5]. In consequence, [10] proposed a new methodology to produce hard negative texts, leveraging modern Large Language Models (LLM), adversarial refinement and human annotation.”
> >
> > **2. “Line 71-72: Winoground is not the only dataset for bidirectional retrieval.”**
> >
> > We plan to rewrite that paragraph:
> >
> > “In a similar fashion, Winoground [9] also includes both retrieval directions to measure model performance. However, as shown by [7], Winoground has several problems: (i) it is very small (it contains 400 instances), which makes the comparison among models difficult, (ii) it contains very few instances which actually measure Vision-Language Compositionality (only 171), and (iii) some other challenges like commonsense reasoning or locating small, out-of-focus objects in low resolution images are very important to perform well on the task. In this work, mimicking Winoground, we also build every instance of the dataset with two images and two captions, but we only focus on compositionality and we provide almost 3 thousand instances, being much larger than Winoground.
> >
> > Recently, two other bidirectional datasets have been published: EqBen [1] and Cola [2]. EqBen has been derived from video-text datasets and also offers a set of synthetic images generated with a graphic engine. However, the test set contains low-quality images (e.g. very dark or blurry pictures) [3] and authors have released a higher quality subset of 280 pairs of image-text pairs, which is even smaller than Winoground. Cola, on the other hand, has no natural texts, as the captions are produced using templates, and covers only a small subset of compositionality phenomena, as it focuses on object-attributes and spatial relations.”
> >
> > **3. “Line 252: I believe VisualGPTScore based on BLIP [4] is still the SOTA on SugarCrepe.”**
> >
> > Yes, we checked the paper and VisualGPTScore obtains an average of 95.62 in SugarCrepe. However, to obtain that score, they use cross-validation on test data to tune a debiasing hyperparameter. If they omit that tuning, i.e. if they do not use test data, they obtain a score of 92.18, which is lower than CLIP-TROHN-Text (93.40). In addition to VisualGPTScore, we evaluated VQAScore in SugarCrepe and we found it to be slightly better than our model, achieving a score of 93.72. In consequence, we plan to rewrite the mentioned paragraph as:
> >
> > “To put it into the context of the state-of-the-art, the strongest models for SugarCrepe are VQAScore [3] (93.72), CapPa [5] (92.88) and GPT-4V [8] (92.19). CLIP-TROHN-Text is on par with VQAScore and outperforms CapPa and GPT-4V, showing it is very strong for image-to-text retrieval, despite its smaller size.”
> >
> > ### **Some minor typos:**
> >
> > We will correct the typos (the mention of the size of Winoground included) and we will include Random Chance performance in all the tables.
> >
> > ### **Feedback on dataset:**
> >
> > Regarding the feedback on the dataset, we thank the reviewer for the pointers. We agree that such low-quality samples are rare. We will perform an additional filtering step and perform a new release of the dataset.

---

> > > ### Author Response · Authors · 2024-08-16
> > > **Response to Reviewer 6Z5K (3)**
> > >
> > > ### **Limitations:**
> > >
> > > **“I would like to know if the proposed training method and dataset improve on Winoground, e.g., in Table 3.”**
> > >
> > > We didn’t evaluate our models on Winoground due to the issues mentioned in [7]: According to those authors, Winoground is not suitable to evaluate VL compositionality, which is the main topic of our paper, and thus decided not to report them. The problems identified were: (i) it is very small (it contains 400 instances), which makes the comparison among models difficult, (ii) it contains very few instances which actually measure Vision-Language Compositionality (only 171), and (iii) some other challenges like commonsense reasoning or locating small, out-of-focus objects in low resolution images are very important to perform well on the task.
> > >
> > > But following the advice of the reviewer, in the following table, we show the obtained results of our models and the baseline model for Winoground (400 pairs):
> > > | **Model**      | **I2T** | **T2I** | **Group** |
> > > |----------------|:-------:|:-------:|:---------:|
> > > | CLIP           |  31.25  |  11.00  |    8.75   |
> > > | CLIPTROHN-Text |   30.5  |  13.25  |    8.5    |
> > > | CLIPTROHN-Img  |   32.5  |   14.0  |    9.5    |
> > >
> > > As can be seen, we do improve over the base CLIP model, but the improvements are small. In fact, CLIPTROHN-Text is even slightly worse. According to [7] samples dealing with compositionality are a subset of Winoground. If we evaluate on that subset (171 pairs of image-captions), the improvement is stronger:
> > >
> > > | **Model**      | **I2T** | **T2I** | **Group** |
> > > |----------------|:-------:|:-------:|:---------:|
> > > | CLIP           |  32.16  |  11.70  |    9.36   |
> > > | CLIPTROHN-Text |  33.33  |  12.87  |    9.36   |
> > > | CLIPTROHN-Img  |  38.60  |  14.62  |   11.11   |
> > >
> > > We agree with the reviewer that those results may indicate a weakness of training with synthetic images. However, there are also some other hypothesis: a) Winoground has only samples of the Swap category, which is the hardest one for contrastive models; b) the use of ViT-B/32, with a patch size of 32x32, may make very difficult to represent “small” details. We think a deeper analysis is needed to know what is happening and accordingly, we will update the Limitations section of the paper:
> > >
> > > “BiVLC offers captions only in English. It would be interesting to extend the dataset to other languages, as some recent works in vision-language models are already doing [11, 12, 13]. Moreover, we only trained contrastive models, due to their suitability for image-to-text and text-to-image retrieval tasks and their availability. In the future, generative multimodal models, which we evaluated but not fine-tuned, could also be explored. Indeed, our approach to fine-tune contrastive models with hard negative images also has its limitations: we evaluated the models in Winoground and we saw that improvements are modest. There are several hypotheses to explain those results and one of them points to the effect of using synthetic images. Deeper analyses are needed to elucidate the real effect of synthetic images to train multimodal models. Finally, as we rely on SugarCrepe hard negative captions, we also use the same categories. Adding more diversity by extending BiVLC to other categories could be beneficial.”
> > >
> > > ### **References:**
> > >
> > > [1] Equivariant similarity for vision-language foundation models. Wang et al. ICCV 2023.
> > >
> > > [2] cola: A Benchmark for Compositional Text-to-image Retrieval. Ray et al. NeurIPS 2023.
> > >
> > > [3] Evaluating Text-to-Visual Generation with Image-to-Text Generation. Lin et al. ECCV 2024.
> > >
> > > [4] Revisiting the Role of Language Priors in Vision-Language Models. Lin et al. ICML 2024.
> > >
> > > [5] Image captioners are scalable vision learners too. Tschannen et al. NeurIPS 2024.
> > >
> > > [6] Going Beyond Nouns With Vision & Language Models Using Synthetic Data. Cascante-Bonilla et al. ICCV 2023.
> > >
> > > [7] Why is Winoground Hard? Investigating Failures in Visuolinguistic Compositionality. Diwan et al. EMNLP 2022.
> > >
> > > [8] Gpt-4 technical report. Archiam et al. Arxiv 2023.
> > >
> > > [9] Winoground: Probing vision and language models for visio-linguistic compositionality. Thrush et al. CVPR 2022.
> > >
> > > [10] Sugarcrepe: Fixing hackable benchmarks for vision-language compositionality. Hsieh et al. NeurIPS 2024.
> > >
> > > [11] No filter: Cultural and socioeconomic diversity in contrastive vision-language models. Pouget et al. Arxiv 2024.
> > >
> > > [12] Pali-x: On scaling up a multilingual vision and language model. Chen et al. Arxiv 2023.
> > >
> > > [13] IGLUE: A benchmark for transfer learning across modalities, tasks, and languages. Bugliarello et al. ICML 2022.

---

> > > > ### Comment · Reviewer_6Z5K · 2024-08-26
> > > > **Thank you for the detailed response!**
> > > >
> > > > I appreciate the detailed rebuttal provided by the authors. My concerns have been addressed; the new generative VLMs experiments are incredibly strong and interesting. I would recommend authors to revise the texts for better coverage of related work (as promised in the rebuttal). I have increased my rating from 6 to 7.

---

> > > > > ### Author Response · Authors · 2024-08-27
> > > > > **Thank you for the review!**
> > > > >
> > > > > We thank the reviewer for the positive feedback. We also think that the experiments with generative VLMs are very interesting. We will revise the texts as stated in the rebuttal.

---

### Official Review · Reviewer_k2kk · 2024-07-09
**Useful dataset for multimodal evaluations**

**Rating:** 8
**Confidence:** 4
**Correctness:** Yes
**Clarity:** Yes

**Review:**

The idea of also evaluating text-to-image tasks is logical and yields good insights about current multimodal model performance.
Its validity is supported by the SOTA improvements on a couple of benchmarks.
This dataset will be helpful in further advancement of multimodal models.

**Strengths:**

The idea is simple, yet yields useful insights and improves multimodal performance.

Adding hard negative images, even though they're generated by AI rather than human-crafted, reveals unbalanced performance of multimodal models in the text-to-image direction.

This dataset will be useful to the multimodal research community.

**Additional Feedback:**

Would capitalizing only the first word in each caption sentence cause any issues for other words that should be capitalized (like names of people/places)?

**Documentation:**

Yes

**Ethics:**

The off-the-shelf models used to generate the hard negative images may have ethics issues in their training data (opaque to the research community).

**Limitations:**

Yes

**Opportunities For Improvement:**

An ideal dataset would contain human-crafted hard negative texts and real hard negative images.
Crafting ~1K hard negative texts would be a task doable by skilled human raters.
Obtaining a real hard negative image could be achieved by generating one as the paper describes, then doing a reverse image search to find the closest real image.

**Relation To Prior Work:**

Yes

**Summary And Contributions:**

The paper presents a large vision-language compositionality dataset (BiVLC) that, unlike previous work, contains both image-to-text (given an image, its description, and a hard negative description, choose the description that matches the image) and text-to-image (given an image, its description, and a hard negative image, choose the image that matches the description) tasks.
This is achieved by expanding the existing text-to-image dataset, SugarCrepe, with hard negative images generated from hard negative descriptions using off-the-shelf text-to-image models (manually filtered for quality) .
It then uses this type of data to train a SOTA multimodal model on BiVLC and SugarCrepe benchmarks.

---

> ### Author Response · Authors · 2024-08-16
> **Response to Reviewer k2kk**
>
> **We would like to thank the reviewer for the positive feedback and the provided ideas. We comment on those ideas and answer the questions here:**
>
> **Opportunities For Improvement:
> “An ideal dataset would contain human-crafted hard negative texts and real hard negative images.”**
>
> This proposal is interesting, but there are also some issues: a) due to the manual efforts needed (craft hard negative captions, verify their quality, verify whether the generated images match the captions, verify whether the natural images mined from the synthetic ones match the captions), we estimate that the cost of the dataset will be significantly higher than ours; b) given the filtering steps are necessary (take into account that in our case, we discarded the 61% of the generated samples), if we craft 1k captions, we may end up with a very small dataset; c) it is not clear whether the evaluation results and conclusion will be very different from ours. Overall, we think it’s a good idea for follow-up work and we plan to check it in the future.
>
> **Additional Feedback:
> “Would capitalizing only the first word in each caption sentence cause any issues for other words that should be capitalized (like names of people/places)?”**
>
> We took that decision because we are using COCO captions, where people/places names are very rare. We performed a manual check of our captions and we found that only 40 out of 2,933 captions contain names that should be capitalized (e.g. Wii or Nintendo). Those represent only 1.36% of the captions in the dataset, so their influence is negligible. Moreover, all the models reported in the paper use tokenizers that lower-case the input texts, so there should not be any difference in the results. Nevertheless, we will revise the dataset and provide a capitalized version.

---

### Official Review · Reviewer_oFAf · 2024-07-26
**A benchmark for evaluating t2i/i2t retrieval with positive & negative pairs**

**Rating:** 5
**Confidence:** 3
**Correctness:** Generally yes.

**Review:**

Pros:

- Contrastive analysis with both positive and negative pairs. This is an interesting point. This work proposes a dataset for checking the retrieval ability of VLMs using both positive and negative image-text pairs. The negative images are generated with diffusion models and then checked by human annotators.
- Both text-to-image and image-to-text evaluation are performed. This is more comprehensive for the experimental analysis.
- Some useful and non-traditional findings are provided in the experiments.

Cons:

- Novelty and contribution. The major novelty is to extend SUGARCREPE and create negative image samples from negative descriptions with image generation model. This may not be sufficient as the core contribution.
- The writing is not satisfying. See the "clarity" comments below.

**Strengths:**

Please see the pros above.

**Additional Feedback:**

None.

**Clarity:**

The writing is generally acceptable, but not very satisfying.

- For example, in the last paragraph of Sec. 1, the authors summarizes a few findings from the experimental analysis on the proposed benchmarks, but seems mistakenly write them as "contributions" rather than "findings".
- Also, the authors seems to provide a model, THORN, but not mentioning it at all in the introduction.

**Documentation:**

A documentation and a URL to access the dataset is provided, though some details may be added to the documentation to it more comprehensive.

**Ethics:**

No ethical concerns, as this work is majorly extended from an already published work (SUGARCREPE dataset).

**Limitations:**

Please see the cons above.

**Opportunities For Improvement:**

Please see the cons above.

**Relation To Prior Work:**

Generally yes. Important related works, like ARO, are discussed.

**Summary And Contributions:**

Summary:

- A benchmark constructed using image generation models for checking text-to-image and image-to-text retrieval.
- Some analysis over existing methods, and a few findings regarding the capability of current models.

---

> ### Author Response · Authors · 2024-08-16
> **Response to Reviewer oFAf**
>
> **We thank the reviewer for the provided feedback. We answer here the commented issues:**
>
> **The topic of the paper:
> “... This work proposes a dataset for checking the retrieval ability of VLMs using both positive and negative image-text pairs. …”**
>
> The paper is about evaluating the compositionality capabilities of visio-linguistic (VL) models, not about retrieval abilities.
>
> **Cons:
> “Novelty and contribution. The major novelty is to extend SUGARCREPE and create negative image samples from negative descriptions with image generation model. This may not be sufficient as the core contribution.”**
>
> Given the limitations of current datasets such as SugarCrepe, VL compositionality has been mainly evaluated and analyzed on the image-to-text retrieval direction only, which gives a limited view of the problem. In fact, our methodology and resulting dataset shows that different conclusions are obtained when considering both image-to-text and text-to-image retrieval directions. The insights we present are novel and interesting for the community, and can be only obtained thanks to our dataset. Besides, we also contribute two new training datasets to fine-tune VLMs for compositionality. For those reasons, reducing the novelty and contribution of our work to “just extending SugarCrepe” is very vague and we kindly ask the reviewer to reconsider this point, and also take into account the significance and quality of our work.
>
> **Clarity:
> “For example, in the last paragraph of Sec. 1, the authors summarizes a few findings from the experimental analysis on the proposed benchmarks, but seems mistakenly write them as "contributions" rather than "findings".”**
>
>
> We will change the writing and distinguish between contributions and findings, as suggested by the reviewer.
>
>
> **“Also, the authors seems to provide a model, THORN, but not mentioning it at all in the introduction.”**
>
> We mention the model (lines 48-49), but we decided to omit the name for several reasons: 1) we think that the idea (“Training with hard negative images”) is more important than the actual implementation that we did (which we call TROHN-img); 2) mentioning a name could give the false impression that the paper is about this model, not about the dataset. We don’t think this and the previous point are enough to judge the clarity of our paper as insufficient.
>
>
> **Documentation:
> “A documentation and a URL to access the dataset is provided, though some details may be added to the documentation to it more comprehensive.”**
>
>
> In our opinion the documentation is comprehensive enough. Please let us know which details are worth being added.

---

### Decision · Program_Chairs · 2024-09-26

**Decision:**

Accept (Poster)

**Comment:**

Overall, the paper has been lauded for the original idea of adding generative AI images to create a more comprehensive  multimodal dataset.

While the writing has been pointed as an issue, the authors have committed to improving it in the rebuttal as well as addressed the reviewers pain points. I am not considering the third review since it is lacking in details as the other two reviews. The dataset is also hosted on hugging face and is of good quality.

Based on the above I am recommending an Accept as a Poster